# Mechanical Performance of Glass-Based Geopolymer Matrix Composites Reinforced with Cellulose Fibers

**DOI:** 10.3390/ma11122395

**Published:** 2018-11-28

**Authors:** Gianmarco Taveri, Enrico Bernardo, Ivo Dlouhy

**Affiliations:** 1Institute of Physics of Materials, Czech Academy of Science, Žižkova 22, 61662 Brno, Czech Republic; idlouhy@ipm.cz; 2Department of Industrial Engineering, University of Padova, 35131 Padova, Italy; enrico.bernardo@unipd.it

**Keywords:** geopolymer composite, wastes incorporation, cellulose fibers, cellulose modification

## Abstract

Glass-based geopolymers, incorporating fly ash and borosilicate glass, were processed in conditions of high alkalinity (NaOH 10–13 M). Different formulations (fly ash and borosilicate in mixtures of 70–30 wt% and 30–70 wt%, respectively) and physical conditions (soaking time and relative humidity) were adopted. Flexural strength and fracture toughness were assessed for samples processed in optimized conditions by three-point bending and chevron notch testing, respectively. SEM was used to evaluate the fracture micromechanisms. Results showed that the geopolymerization efficiency is strongly influenced by the SiO_2_/Al_2_O_3_ ratio and the curing conditions, especially the air humidity. The mechanical performances of the geopolymer samples were compared with those of cellulose fiber–geopolymer matrix composites with different fiber contents (1 wt%, 2 wt%, and 3 wt%). The composites exhibited higher strength and fracture resilience, with the maximum effect observed for the fiber content of 2 wt%. A chemical modification of the cellulose fiber surface was also observed.

## 1. Introduction

Geopolymers and alkali-activated materials (AAMs) are considered as the cementitious materials of the future [1], to be applied mainly in building and civil infrastructures [2,3,4]. What makes these materials widely attractive is the low CO_2_ emission process of production, coupled with mechanical properties at least comparable to Portland cement (OPC) [5,6,7]. To date, however, extensive market diffusion has failed due to several reasons, including the cost of production, upscaling, and standardization of the process [8]. Nevertheless, so far, no other materials have been found to be more suitable than geopolymers and AAMs for facing the constantly increasing concern regarding climate change due to greenhouse gas emissions in the atmosphere, with 8% of the annual CO_2_ emissions being accounted for by OPC production [1,7,9].

To decrease the cost of production, a fundamental solution consists of the incorporation of aluminosilicate waste, such as fly ash (a byproduct of coal combustion in thermal power plants) [10,11,12,13,14,15,16], as raw materials. The alkali-activation of fly ash induces the formation of a Ca-modified sodium aluminosilicate hydrates (N-A-S-H) gel through the polycondensation of aluminosilicate–lime species in a semiamorphous network composed of long-chain molecules [9,17,18]. In contrast to the benefits of its low cost and versatility of production, the extensive presence of hydrate groups in the chemistry of fly ash-based AAMs does not make this material more durable than OPC, unlike geopolymers [19,20,21,22]. The latter materials are characterized by a low Ca content (which normally favors the formation of a calcium silicate hydrates gel, CߝSߝH) and a silica-to-alumina ratio (SiO_2_/Al_2_O_3_) exceeding 2, and thus they yield a semiamorphous three-dimensional and highly cross-linked aluminosilicate microstructure with a much lower amount of hydrate groups than in AAMs [21,23,24]. The additional supply of reactive silica could be provided by introducing additional waste-derived raw materials, such as recycled glass (from urban and industrial waste collection), with the obvious advantages of cost reduction and waste management [16,25,26,27,28]. Among all the possible variants of recycled glass utilized in geopolymerization, borosilicate glass (BSG) cullet from dismantled or discarded pharmaceutical vials is an intriguing alternative [29], since it was demonstrated that it also provides reactive borates in polycondensation, replacing alumina in its role in geopolymerization [30].

Despite all these recent developments, geopolymers still suffer from sudden unstable fractures due to their extreme brittleness [31,32]. The excessive low resistance to crack initiation/propagation is not due only to the fragility of the geopolymer product of reaction, but also to the extent of porosity and crack production during hardening. The amount of macrodefects generated by the process could be controlled to a limited extent through chemical (e.g., alkalinity, liquid-to-solid ratio) and physical conditions (e.g., humidity) in curing [33,34,35]. Alternatively, the production of composites from a geopolymeric matrix was extensively investigated as an effective solution to increase the fracture toughness, due to the mechanisms of pull-out and crack bridging of dispersed fibers [36,37,38,39,40]. Above all, cellulose seems to be suitable for geopolymer composites due to its chemical stability and specific tensile strength [41,42,43]. 

Here, a comparative study carried out on the effects of chemical (alkalinity and silica-to-alumina ratio) and physical parameters (relative humidity and soaking time) on the mechanical and microstructural properties of geopolymers is reported. Geopolymer composites were also produced by dispersing cellulose fibers, and the effect of fiber content was assessed in terms of bending strength, fracture toughness (chevron notch test), and fracture micromechanisms. 

## 2. Materials and Methods 

### 2.1. Manufacturing of Geopolymer Samples

Fly ash, a coal combustion byproduct, from a Bohemian thermal power plant (Počerady power plant, North Bohemia, Czech Republic) was used as a primary aluminosilicate source. The fly ash was dry-mixed with borosilicate glass from recycled pharmaceutical vial cullet (Kimble/Kontes, Vineland, NJ, USA) and activated with a caustic soda solution, prepared by dissolving NaOH pellets (American Ceramic Society reagent, 97%, pellets) purchased from Sigma Aldrich (Saint Louis, MO, USA) in distilled water. The geopolymer samples were manufactured according to 6 different formulations, in which the powder mixture (SiO_2_/Al_2_O_3_ ratio), the molarity of the activator, and the curing time were modulated in order to investigate the influence of these parameters on geopolymerization. A schematic summary of the methodology is reported in Table 1. In batches ‘Mix-1’ to ‘Mix-4’, samples were prepared by activating dry mixes of fly ash from 70 wt% to 30 wt% and BSG from 30 wt% to 70 wt% in a 13 M NaOH solution and cured for 1 day. The samples processed through the ‘Mix-5’ batch were based on the ‘Mix-2’ formulation and cured for 3 days, and finally, ‘Mix-6’ samples were activated using a 10 M NaOH solution. The alkali solution was added in a sufficient amount to ensure workability to the slurry. In all the mixes, the liquid-to-solid ratio ranged between 0.4 and 0.5. The obtained slurry was cast in rubber molds and cured at 85 °C. The humidity in the samples was retained in two different ways [35]:Method 1: Molds were sealed in latex bags.Method 2: Molds were closed in an air-tightened jar with some water.

Irrespective of the methodologies of processing, after curing, the samples were demolded and exposed to air for one week prior to testing in order to complete the geopolymerization. 

The relative density of the samples in all the cases was calculated to be around 70%, according to the Archimedes method (theoretical density of 2.08 g/cm^3^ and real density of 1.44 g/cm^3^ on average), using high-purity water as a buoyant. The weighting of the specimens was carried out using a Denver analytical balance (±0.0001 g precision). 

### 2.2. Manufacturing of Geopolymer Composite Samples

Mix-1 was properly modified in order to combine the dry mix with cellulose fibers in different percentages (from 1 to 3 wt%); the according weight content substitute was fly ash (from 69 wt% to 67 wt%). The reduction of the fly ash content was not considered sufficient to effect significant change in the properties of the material. The cellulose fibers were provided by CIUR A.s. (Brandýs nad Labem, Czech Republic). The dry mix was then diluted in distilled water and sonicated for one hour to guarantee homogenization of the mix and to unravel the bundles of cellulose fibers. The suspension was dried overnight and the dry mix was activated in a 13 M NaOH solution, forming a slurry whose liquid-to-solid ratio was between 0.8 and 0.9. The slurry was cast according to method 1 and cured at 85 °C for one day. As mentioned previously, the samples were demolded after curing and kept in air for one week prior to testing. 

### 2.3. Mechanical Testing

Flexural strength and fracture toughness were determined through a 3-point bending test and chevron notch (CVN) test, respectively, conducted on 3 × 4 × 16 mm specimens using a ZWICK Z50 screw-driven machine (Ulm, Germany). The CVN test was performed in a 3-point bending configuration on specimens with a notch depth of 1 mm. The value of the fracture toughness was calculated from the ultimate bending load according to the following formula:(1)$$\;K_{IC}=\frac{\;Y_{min}^{*}\cdot \;F}{B\;\cdot \;W^{\frac{1}{2}}},$$
where *F* is the measured bending load corresponding to unstable crack development, $Y_{min}^{*}$ is the minimum of the geometry function $Y^{*}$ which is dependent on the notch depth *a*_0_ and the geometry, *B* is the thickness, and *W* is the width of the specimen. The reliability of the chevron notch technique for fracture toughness determination of fiber-reinforced brittle matrix composites has been probed elsewhere [44]. The data elaboration was carried out using interquartile range (IQR) () statistics with a whisker factor of 1.5 [45,46].

### 2.4. Microstructural Investigation

The microstructure was investigated through SEM microscopy with a Tescan LYRA 3 XMH FEG equipped with an X-Max80 Energy-dispersive X-ray spectroscopy (EDS) detector for X-ray chemical analysis. The chemical composition of the raw materials was analyzed through X-ray fluorescence (XRF) analysis using a RIGAKU (Tokio, Japan) ZSX100e model operating at 60 kV and 150 mA and equipped with Wavelength-dispersion X-ray spectroscopy (WDS), an X-ray Rh tube working at 3 kW, a scintillation counter for heavy element detection, and a gas-flow proportional counter (Ar–methane 10 %) for the detection of light elements.

## 3. Results and Discussion

### 3.1. Geopolymer Characterization

The geopolymerization process was tuned by considering the influence of the SiO_2_/Al_2_O_3_ ratio, the molarity of the alkali solution, and the soaking time and humidity during curing, although the parameters influencing the process also include the temperature of curing, nature of the alkali solution, silica-to-alkali ratio, liquid-to-solid ratio, and so forth. The calculation of the SiO_2_/Al_2_O_3_ ratio was based on the chemical composition of the raw materials, that is, fly ash and BSG glass. Table 2 reports the chemical composition of both the raw materials. 

The four fly ash/BSG formulations (from Mix-1 to Mix-4), as described in the experimental section, are characterized by a silica-to-alumina ratio of 2.7, 3.3, 4.2, and 5.0, respectively (Table 1). Considering other ratios as well (silica-to-soda and water-to-soda), all the formulations fall in the chemical ranges giving a stoichiometric geopolymer [23,24]. Interestingly, it was demonstrated in previous studies that the utilization of BSG glass as a silicate supplier of the geopolymerization also provides, during the dissolution stage, borate species, which eventually become part of the final microstructure as additional building blocks, giving rise to a B–Al–Si network [30]. Boron oxide, similarly to alumina, can be found in nature as diboron trioxide (B_2_O_3_), and dissolved borates can display a trigonal planar (BO_3_) or a tetrahedral (BO_4_) configuration (3-fold and 4-fold configurations, respectively). During geopolymerization, aluminates and borates concurrently rearrange in a 4-fold configuration (with charge compensation provided by alkali ions) to condense with tetrahedral silicates and to form the boron–alumino–silicate chains [30,47]. Due to this reason, a more refined formula of the previous ratio is provided, also taking into account the borates’ contribution:(2)$$\alpha =\;\frac{\%\mathrm{Si}O_{2}}{\%\left(Al_{2}O_{3}+B_{2}O_{3}\right)},$$

For the calculation of the *α*-coefficient, we employed the weight percentages given by XRF analysis (Table 2); the *α*-coefficients for Mix-1, Mix-2, Mix-3, and Mix-4 were calculated to be 2.2, 2.5, 2.8, and 3.0, respectively (see Table 1).

Figure 1a reports the flexural strength values of the whole set of the tested specimens in three-point bending. The results of the three-point bending test revealed an increase of the flexural resistance of the geopolymers up to 50% if the *α*-coefficient is increased from 2.2 to 2.5. The higher amount of silicates in the formulation, indeed, may help the formation of the more cross-linked three-dimensional network, providing higher strength to the geopolymer. The maximum flexural strength values were observed to decrease if the *α*-coefficient was increased from 2.5 to 3.0, whereas the median value of the IQR whiskers box experienced a continuous increase and the data dispersion became narrower (Mix-3 and Mix-4 in Figure 1a). The reason of this trend was attributed to a deterioration of the microstructure, giving rise to a lowering of the maximum flexural strength value and to a decrease of the porosity and crack formation, inducing a refinement of the statistic and the average flexural strength values. The macrodefects’ formation in the geopolymer samples could be unrelated to the changes in the chemistry of the system.

A similar trend in the flexural strength is also associated with the higher molarity of the alkali solution: Figure 1b demonstrates that an increase in molarity of the NaOH solution from 10 M to 13 M leads to an increase of the flexural strength of geopolymer samples from 4 MPa (Mix-5) to almost 14 MPa (Mix-2), and to a higher data dispersion. This result is discordant with those of several prior studies, asserting that a higher amount of alkali cations hinders the condensation of long polymeric chains, which generally gives less microstructural stability to the geopolymeric compound [48]. The microstructural refinement is not always synonymous with the mechanical optimization of the material, as it has to be appropriately weighted to the efficiency of other stages of the geopolymerization, which in turn depends on other factors. One of these is the initial stage of dissolution, where the building blocks of the polymeric structure, namely borates, aluminates, and silicates, are provided from. The dissolution of the ionic species is strictly related to the raw materials’ nature and the concentration of the alkali solvent. Specifically, a higher concentration of the alkali activator yields to a better dissolution of the boron–alumino–silicate sources, improving the degree of reaction [49,50].

The results from the three-point bending test conducted on Mix-6 samples showed that an increased duration of exposure to the curing temperature decreases the flexural strength of the material by up to half of the average flexural strength (Figure 1b). This drop in the bending properties could be attributed to multiple reasons, but in general, an increase in the temperature exposure can induce a larger shrinkage and more substantial crack formation. A comparative study between the microstructures of the geopolymer samples tested in bending is not reported in this work, as the samples did not show any significant differences. Evidence of the microstructure is reported in Figure 2a, in which porosity and cracks are homogenously present everywhere, reporting a characteristic morphology associated with hardening and shrinkage during the curing of geopolymers featuring a high SiO_2_/Al_2_O_3_ ratio [12,17,51]. The amount of porosity and defects is inferable from the calculated relative density (see the Materials and Methods section).

In all the cases, the Mix-2 samples provide the higher bending strength, but at the same time, larger data scattering. This is also attributed to porosity and crack formation, which are strictly related to the amount of water in the slurry and to the methodology of sealing and amount of humidity in the bag. For instance, a few percentages of humidity change can induce a significant variation in the mechanical properties of the geopolymers [33].

The parametrical study of the geopolymerization as a function of humidity goes beyond the aim of this work, but two sealing methodologies of the samples in the curing phase were tested. Method 1 involves humidity only by water evaporation of the slurry, while method 2 provides more water to the system. The flexural results related to Figure 1 are attributed to samples cured according to method 1. The data testing of the samples processed according to method 2 are not reported, as they do not show any appreciable bending resistance. The microstructure of these samples was observed by SEM microscopy and compared to the Mix-2 sample. Figure 2b revealed a deficiency in the degree of reaction and a much lower amount of geopolymeric product in samples processed by method 2 (Figure 2b) as compared to method 1 (Figure 2a), confirming that the supply of extra water in the curing stage hinders the geopolymerization reaction [33,35].

### 3.2. Geopolymer Composites

The comparative study of the fracture performance of geopolymer samples, albeit tested, were not reported, as the CVN test gave values of fracture toughness all below 0.3 MPa·m^1/2^. The resistance to the crack propagation was significantly improved with the addition of 1 wt%, 2 wt%, and 3 wt% cellulose fibers into the geopolymeric matrix (see Materials and Methods section). Besides the fracture resistance, the flexural strength similarly improved, as depicted in Figure 3. In all the three composite samples, the flexural strength improved as compared to the plain geopolymer samples. The highest flexural strength was observed with 2 wt% cellulose dispersed throughout the sample, with an average value of 18 MPa. This value is three-fold higher than the ‘Mix-1’ value (see Figure 1a), and higher than the flexural strength reported in the literature for an analogous fly ash-based geopolymer matrix composite with dispersed cellulose fibers [41]. Geopolymeric composite samples with a cellulose content exceeding 2 wt% induce a reduction of the flexural strength (Figure 3a), attributed to excessive Na^+^ absorption by the cellulose fibers. The same trend was also observed for the fracture toughness, reported in Figure 3b, in which the highest fracture resistance (0.6 MPa∙m^1/2^) is displayed by the 2 wt% sample, with an average increase of the fracture toughness of up to 300% as compared to ‘Mix-1’ sample. The value of the fracture toughness is in line with the values reported in the literature [41].

The absorption of alkali cations (especially Na^+^) by the cellulose fibers is a widely known process known as ‘mercerization’. The cellulose swells due to the interference of alkali cations between the polymeric chains, creating electrostatic layers of positively charged Na^+^ ions and negatively charged polymeric chains [52]. What is not equally reported is the absorption of other chemical elements, which creates, in cellulose-based composites, a sort of transitional interface between the inner part of the fiber and the matrix. Tonoli et al. [53] demonstrated the formation of a modified surface on cellulose fibers dispersed in the cement matrix, due to Ca^2+^ absorption from the C–S–H network of the cement. Previously, Merrill et al. [54] published a study reporting the sodium silicate sorption in cellulose fibers, in which it was proved that the cellulose can also absorb silicon ions from the water glass, although to a lower extent than sodium, involving a polycondensation with the polymeric chains.

In our geopolymer matrix samples, irrespective of the weight amount of the dispersed fibers, cellulose was found to undertake a surface modification and a new phase formation at the interface with the geopolymeric matrix. Figure 4a shows detail of a cellulose fiber, whose surface is clearly modified. It is evident from the figure that the affected polymeric fibrils at the fiber surface are rearranged in an orthogonal configuration to the geopolymer interface (see insert in Figure 4a), whereas at the core, the fibrils are parallel to the interface line. EDS analysis was conducted to detect the chemical species absorbed by the new phase, and the results are reported in Figure 4b. The chemical analysis was carried out in three different zones in order to draw a conclusion regarding the elements’ absorption: zone 1—the cellulose fiber core, zone 2—the cellulose/geopolymer interface, and zone 3—the geopolymeric matrix. In all zones, the amount of sodium was the same, whereas the silicon was observed to be more abundant in zone 3 than zone 2, as expected. What was, however, unexpected was a deficiency of oxygen in the modified interface (zone 2), suggesting a condensation between the hydroxyl groups of the polymeric chains of the cellulose and geopolymer with corresponding water release. However, this hypothesis should be supported by further spectroscopic analysis, unequivocally demonstrating the formation of a new phase formed by organic–inorganic polymeric chains.

The presence of the new phase between the two organic–inorganic compounds can influence the mechanisms of fracture of the composite samples. The observation of the fracture surface (Figure 5a,b) and the polished surface (Figure 5c,d) of the geopolymer matrix samples revealed the presence of both the mechanisms of pull-out and crack bridging. For instance, the investigation of the fracture surface clearly evidenced that during fracture, the fibers were exposed to pull-out mechanisms from the geopolymeric matrix (Figure 5a). This was especially highlighted by observations of delamination sites, such as the one reported in Figure 5b. The mechanisms of crack bridging were rather evident on the polished surface (Figure 5c). Due to the presence of the fiber, the crack was both hindered and deflected from the direction of propagation. However, the formation of the new phase induces also mechanisms of fiber tearing, as shown in Figure 5d. Indeed, in some cases, the interface strength may be higher than the resistance of the single compound, causing fiber failure instead of fiber delamination. Figure 5b shows the remainder of a cellulose fiber in the delaminated site, suggesting an inhomogeneous nature of this phase. Therefore, to understand the whole dynamics of failure in the geopolymer matrix composite, a detailed investigation of the chemical nature and mechanical performance of the new phase is required and will be the object of future work.

## 4. Conclusions

In summary:Geopolymer samples incorporating fly ash and borosilicate wastes were processed using four different formulations (Mix-1, Mix-2, Mix-3, and Mix-4), two alkali solution molarities (10–13 M, Mix-5), and curing times (1–3 h, Mix-6) in order to evaluate the influence of these chemical/physical parameters on geopolymerization.The rate of influence was assessed in terms of flexural strength. The results of the bending test revealed that the higher flexural strength (13 MPa) was associated with the Mix-2 sample, which was activated with a 13 M NaOH solution and cured for 1 day.The fracture toughness of the geopolymer was enhanced by up to 4 times when geopolymer matrix samples including 2 wt% of cellulose fiber were processed.A mercerization and a surface modification of the cellulose fibers were also observed, and EDS analysis suggested a possible chemical interaction between the organic and inorganic polymeric chains.The dynamics of failure of the geopolymer matrix composite includes crack bridging, fiber pull-out, and fiber-tearing mechanisms.

## Figures and Tables

**Figure 1 materials-11-02395-f001:**
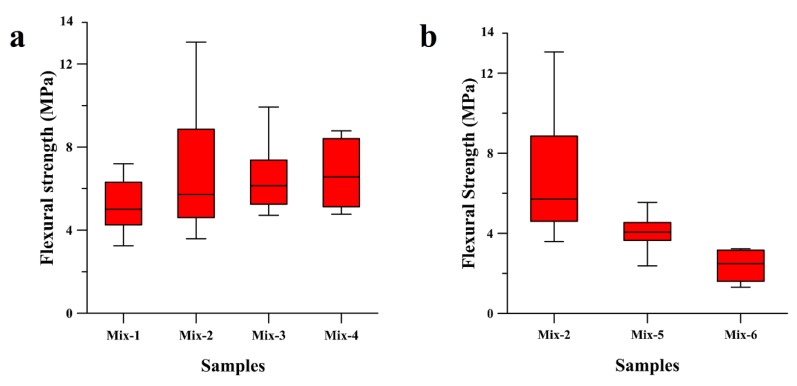
Comparative study of the flexural strength of geopolymer samples in terms of (**a**) silica-to-alumina ratio of the mixtures from Mix-1 to Mix-4 (2.7, 3.3, 4.2, 5.0 respectively) and (**b**) molarity of the activator and curing time used for Mix-2 (13 M for 1 day), Mix-5 (13 M for 3 days) and Mix-6 (10 M for 1 day).

**Figure 2 materials-11-02395-f002:**
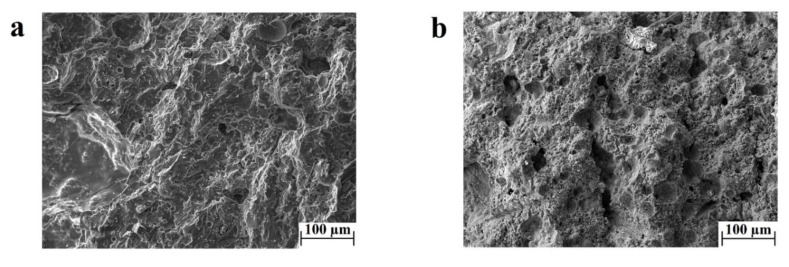
SEM images of (**a**) the ‘Mix-2’ sample sealed according to method 1 and (**b**) the ‘Mix-2’ sample sealed according to method 2.

**Figure 3 materials-11-02395-f003:**
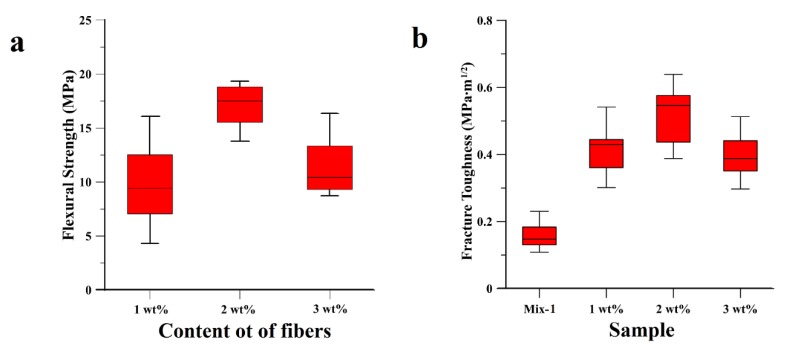
Comparative study of the mechanical and fracture performances of the cellulose fibers geopolymer matrix composites in terms of (**a**) Flexural strength of samples with 1 wt%, 2 wt% and 3 wt% of dispersed cellulose fibers and (**b**) fracture toughness of the geopolymer sample (Mix-1 formulation) and geopolymer matrix composite samples (1 wt%, 2 wt% and 3 wt%).

**Figure 4 materials-11-02395-f004:**
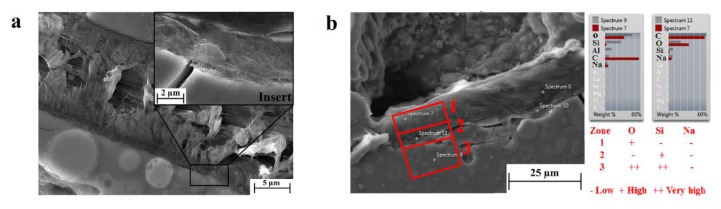
(**a**) SEM image of a cellulose–geopolymer interface, including a close-up of the new phase (insert), and (**b**) Energy-dispersed X-ray spectroscopy (EDS) analysis of the three zones of interest.

**Figure 5 materials-11-02395-f005:**
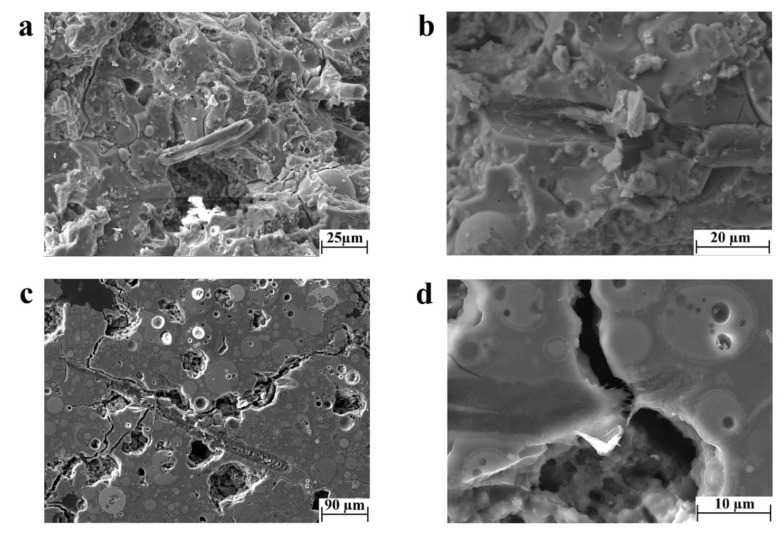
Fracture and polished surface observations of the microstructure of the geopolymer matrix composite through SEM images of (**a**) fiber pull-out and (**b**) negative site of a delaminated fiber, (**c**) crack-bridging, and (**d**) fiber failure.

**Table 1 materials-11-02395-t001:** Summary of the methodology of preparation of the fly ash (FA) and borosilicate (BSG) based geopolymer samples and related parameters.

Sample	Powder Mixture (FA%–BSG%)	Molarity (M)	Curing Time (day)	SiO_2_/Al_2_O_3_	α Coefficient
Mix-1	70–30	13	1	2.7	2.2
Mix-2	55–45	13	1	3.3	2.5
Mix-3	40–60	13	1	4.2	2.8
Mix-4	30–70	13	1	5.0	3.0
Mix-5	55–45	13	3	3.3	2.5
Mix-6	55–45	10	1	3.3	2.5

**Table 2 materials-11-02395-t002:** Chemical compositions of raw fly ash and recycled BSG glass powders.

	SiO_2_	Al_2_O_3_	B_2_O_3_	Fe_2_O_3_	CaO	K_2_O	Na_2_O	LOI ^1^	Remainder
**Fly ash (wt%)**	46.3	26	/	13.9	3.5	3.9	0.2	0.7	5.5
**BSG (wt%)**	72	7	12	/	1	2	6	/	/

^1^ Loss of ignition (LOI)—calculated through weight loss after fly ash firing.

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
