# Peer review of "Mechanical Performance of Glass-Based Geopolymer Matrix Composites Reinforced with Cellulose Fibers"

_materials, 2018, doi:10.3390/ma11122395_

Reviewer 1 Report

The paper aims at investigating the effect of cellulose fibers content in terms of bending strength, fracture toughness (chevron notch test) and fracture micro-mechanisms in glass-based geopolymers, incorporating fly-ash and borosilicate glass.

The paper reports new knowledge and is of interest to readers. Even if the mechanical characterization seems to be well performed, the main drawback of this work, in my point of view, is the lack of chemical characterization of the geocomposite products.

This additional measurement would give a better insight to the interaction (or lack of interaction) between the geopolymeric binder and fibers. Additionally, the preparation procedure is very confusing and a more detailed mix design of the glass-based geopolymer is suggested.

So, I recommend this manuscript for publication after the above mentioned major revisions.

Author Response

Dear Reviewer,

Thank you for the precious remarks you have proposed for this manuscript. They guided us to a careful revision of the paper.  Below we report the answers to your queries:

- The chemical characterization of the geopolymeric product and the interphase between geopolymer matrix and cellulose fibers is of prominent importance to define the interaction between matrix and filler, as your correctly reported in your comments. Despite the usefulness of these investigations, our intention is to incorporate this information in the following study, so as to give a continuous and complete overview of the topic to the readers. Moreover, a whole chemical characterization of the single compounds requires a long time (half a year) to be completed. Anyway, we specified in the text that it will be done in the upcoming publication.

- Your comment was appropriate, therefore we changed the pattern of the samples methodology in a clearer way, by modifying the samples designation and inserting a summary table (Table 1), and editing the Figure 1 accordingly as well. 

Reviewer 2 Report

The authors are recommended to design more experiments to show more data points for alfa coefficient dependence to improve flexural strength. Otherwise, the conclusion is a stretch.

Author Response

Dear Reviewer,

Thank you a lot for the positive review and the precious remarks you have proposed for this manuscript. They guided us to a careful revision of the paper. Below the answer to your comment:

- we totally agree with this comment and we also believed that more insights were necessary to better define the influence of the alpha coefficient. Thus we carried out further flexural strength evaluations on two other sample batches with higher content of silica (higher alpha coefficient) which led us to a better understanding of the trend. 

Round  2

Reviewer 1 Report

The authors provided all the necessary changes in the manuscript. So I recommend it for publication.